# cDNA Transcriptome of *Arabidopsis* Reveals Various Defense Priming Induced by a Broad-Spectrum Biocontrol Agent *Burkholderia* sp. SSG

**DOI:** 10.3390/ijms23063151

**Published:** 2022-03-15

**Authors:** Ping Kong, Xiaoping Li, Fred Gouker, Chuanxue Hong

**Affiliations:** 1Virginia Tech, Hampton Roads Agricultural Research and Extension Center, 1444 Diamond Springs Road, Virginia Beach, VA 23455, USA; lixiaopi@vt.edu (X.L.); chhong2@vt.edu (C.H.); 2United States Department of Agriculture, Agricultural Research Service, Floral and Nursery Plants Research, Beltsville, MD 20705, USA; fred.gouker@usda.gov

**Keywords:** *Arabidopsis*, cDNA transcriptome, Oxford Nanopore Technology (ONT) sequencing, defense priming, induced systemic resistance (ISR), systemic acquired resistance (SAR), biocontrol agent, leaf endophyte, *Burkholderia* sp.

## Abstract

*Burkholderia* sp. SSG is a potent biological control agent. Even though its survival on the leaf surface declined rapidly, SSG provided extended, moderate plant protection from a broad spectrum of pathogens. This study used *Arabidopsis* Col-0 and its mutants, *eds*16-1, *npr*1-1, and *pad*4-1 as model plants and compared treated plants with non-treated controls to elucidate whether SSG triggers plant defense priming. Only *eds*16-1 leaves with SSG became purplish, suggesting the involvement of salicylic acid (SA) in SSG-induced priming. cDNA sequencing of *Col*-0 plants and differential gene expression analysis identified 120 and 119 differentially expressed genes (DEGs) at 6- and 24-h post-treatment (hpt) with SSG, respectively. Most of these DEGs encoded responses to biotic and abiotic stimuli or stresses; four DEGs had more than two isoforms. A total of 23 DEGs were shared at 6 and 24 hpt, showing four regulation patterns. Functional categorization of these shared DEGs, and 44 very significantly upregulated DEGs revealed that SSG triggered various defense priming mechanisms, including responses to phosphate or iron deficiency, modulation of defense-linked SA, jasmonic acid, ethylene, and abscisic acid pathways, defense-related gene regulation, and chromatin modification. These data support that SSG is an induced systemic resistance (ISR) trigger conferring plant protection upon pathogen encounter.

## 1. Introduction

Plant diseases are significant threats to food security, and they are also a burden on the global economy [1,2]. With increasing concerns on the cost, safety, fungicide resistance, and environmental footprints of chemical control in plant disease management [3], biological control becomes a promising alternative because it uses beneficial microorganisms that have opposite features and multiple modes of action [4,5].

Beneficial microbes can directly kill or inhibit pathogens with antibiotic metabolites, hydrolytic enzymes, siderophores and quorum quenching, nutrient, and space competition. They can also suppress pathogens by preparing and improving the defense capacity of plants or defense priming [6,7,8]. The latter modes of action focus on modulating the plant’s immune system and hormone levels. Similar to plant pathogens, beneficial microbes can produce microbe-associated molecular patterns (MAMPs) and interact with plants leading to pattern-triggered immunity (PTI) [9,10]. The identified MAMPs include flagellin, lipopolysaccharides, glycoproteins, and chitin [6,11]. Beneficial microbes can also produce other elicitors, including siderophores, antibiotics, and low molecular weight volatile compounds derived from different biosynthetic pathways for improved plant immunity [12]. These elicitors trigger the plant defense system at the point of recognition and lead to immunity against pathogens in the whole plant or induced systemic resistance (ISR).

Generally, ISR is different from the systemic acquired resistance (SAR) that is usually triggered by pathogens and mediated by the plant defense hormone salicylic acid (SA)-dependent pathways where SA activates the expression of a set of pathogen-related genes through a redox-regulated protein, Non-expressor of PR genes 1 (NPR1) [13]. Instead, it is usually mediated by an SA-independent pathway where other plant defense hormones jasmonic acid (JA) and ethylene (ET) are central players. Some beneficial microbes are ET regulators by producing 1-aminocyclopropane-1-carboxylate (ACC) deaminase that reduces ET levels while promoting plant growth and defense [14,15]. However, the signaling pathways of ISR can be different depending on the microbial species and the plant species [16,17,18]. The camalexin and glucosinolates mediated SA pathway [19] and a pathway involving both SA and JA/ET signaling via NPR1 functioning in different cellular components [6,20] have been reported for ISR. ISR involving other plant hormone pathways such as abscisic acid (ABA) is also reported [21], although it is often associated with herbivore-induced defense [8].

Both rhizosphere and phyllosphere beneficial microbes have been reported as ISR inducers [6,18,22]. However, to date, most research on beneficial microbe-mediated ISR has focused on rhizosphere microbes. There is increased attention to phyllosphere microbes [5], but *Burkholderia* sp., a new group of promising biocontrol agents [23,24,25], has not been included. We recently isolated an endophytic *Burkholderia* sp., SSG, from boxwood leaves, which suppresses a broad spectrum of plant diseases caused by bacteria, fungi, oomycetes, and viruses while promoting plant growth [26,27,28]. One interesting observation while testing its disease control efficacy was that SSG did not survive more than a week on the boxwood leaf surface [29], but it provided moderate protection to the plants that were challenged with pathogens for more than a week [26,28]. This extended protection implicates interactions between SSG and plants. SSG has a capacity to produce many secondary metabolite clusters and products, including antibiotics, ACC, siderophores, and other molecules that can be used as elicitors for ISR [27,30]. However, whether SSG may trigger defense priming is yet to be elucidated.

Plant transcriptomes have shed light on plant responses to beneficial microbes [18,31,32]. RNA sequencing (RNA-Seq) provides far higher coverage and greater resolution of the dynamic nature of the transcriptome over previous Sanger sequencing- and microarray-based methods [33]. Yet, RNA-Seq for transcriptome analysis relies on known reference sequences that are not available for the plants we have tested with SSG. Therefore, this study used the model plant *Arabidopsis thaliana* Col-0, and its SA mutant plants to elucidate SSG extended plant protection beyond its normal survival time. cDNA of Col-0 treated with and without SSG was sequenced and then compared to determine the transcriptional response to SSG. Meanwhile, morphological responses of all test plants to SSG were also examined. This study provides new insights into plant defense priming triggered by biological control agents and the application of SSG-mediated ISR for plant health enhancement against upcoming pathogens.

## 2. Results

### 2.1. Differential Gene Expression in Col-0 Treated with SSG at Two Sampling Times

EdgeR analysis showed the differential gene expression between plants treated with SSG in Phosphate-Buffered Saline (PBS) and the control plants treated with PBS alone at 6 and 24 h post-treatment (hpt) (Figure 1). The differentially expressed genes (DEGs) were 120 and 119 at 6 and 24 hpt, respectively (Appendix A). Among the 120 DEGs at 6 hpt (Appendix A), 62 were log_2_ FC ≥ 2 (upregulation) while 58 were log_2_ FC ≤ −2 (downregulation). Among the 119 DEGs at 24 hpt, 64 were upregulated, and 55 were downregulated (Appendix A). The upregulated and downregulated DEGs were similar at 6 hpt (Figure 1a) and 24 hpt (Figure 1b), but the upregulated genes were slightly fewer at 6 dpt than at 24 hpt while the downregulated genes were counted opposite.

### 2.2. Gene Ontology (GO) Analysis of DEGs with SSG

Functional categorization of DEGs (Appendix A) indicated that SSG regulated genes involving at least 42 biological processes (BP), 23 cellular components (CC), and 17 molecular functions (MF) terms at 6 hpt. Likewise, the DEGs at 24 hpt involved the same number (42) of BP, 22 CC, and 19 MF (Appendix A).

The GO term enrichment analysis on BP detected more terms: 67 significant GO terms from the 6 hpt DEGs, and 217 significant GO terms from the 24 hpt DEGs (Appendix A). Together, 17 (7.3%) of these terms were uniquely associated with the 6 hpt DEGs, and 167 (71.7%) were uniquely associated with the 24 hpt DEGs, while a total of 49 terms (21%) were shared between the 6 hpt and 24 hpt DEGs. Of those shared BP terms, they were classified into five major functions, including secondary metabolic process, cellular process, regulation of biological quality, and the responses to wounding and stimuli (Appendix A).

The most statistically significant terms of 6 hpt and 24 hpt were related to stress or stimulus responses (Figure 2). The BP terms unique to 6 hpt were in four groups involving the response to light intensity, protein folding, transmembrane transport, and chemical homeostasis (Appendix A). Those unique to 24 hpt were in 18 groups, indicating that extended exposure to SSG resulted in more complex biological processes. The eight major groups involved response to Karrikin, oligosaccharide metabolic process, amine metabolic process, regulation of immune system process, vitamin metabolic process, plant-type cell wall loosening, proximal/distal pattern formation, and glycoside biosynthetic process (Appendix A).

### 2.3. Expression Patterns of DEGs Involving Responses to Biotic and Abiotic Stimuli and Stresses

DEGs in the SSG-treated Col-0 concentrated in BP involving biotic and abiotic stimulus responses or stress (Figure 2) and differed in expression patterns (Figure 3a). More upregulated DEGs encoded responses to biotic stimuli at both time points. A similar expression pattern was found for the genes encoding the external stimuli response, indicating that the plant detected MAMPs from SSG, and the response to biotic stimuli was augmented. A small amount of DEGs encoded the response to light stimuli, and their expression pattern was the opposite to responses to biotic stimuli (Figure 3a), suggesting a negative correlation of plant light sensitivity with the response to biotic stimuli. The percentage of upregulated and downregulated DEGs encoding other stimulus responses were similar, indicating the homeostasis of these processes under the influence of SSG.

DEGs encode responses to biotic and abiotic stimuli and stresses involving a number of CC and MF terms. Among the encoded CC terms, chloroplast was most predominant, followed by cytoplast, cytosol, and nucleus (Figure 3b). For the encoded MF terms, activities of hydrolase, transferase and catalytic, and binding activities of protein and other molecules were predominant (Figure 3c). These abundant components and functions are likely the niche of defense priming triggered by SSG.

### 2.4. Shared DEGs and Statistically Significantly DEGs at 6 and 24 hpt with SSG

Only 10% of DEGs were present at both 6 and 24 hpt. Twenty-three of the 239 DEGs detected were consistently regulated at both sampling points (Table 1). They were responsible for 21% of all the significant BP terms involving responses to wounding and stimuli and secondary metabolic process, cellular process, regulation of biological quality (Appendix A).

Eleven of these DEGs were consistently downregulated or upregulated at both 6 and 24 hpt (Table 1). Specifically, six DEGs were consistently downregulated. These genes mainly encoded transportation, protein modification, responses to ABA, and other abiotic stresses. On the other hand, five DEGs were upregulated at both 6 and 24 hpt. These genes mainly encoded the GO terms such as SA-biosynthesis and signaling and regulation, pathogen defense, and stress response.

The rest of the 12 DEGs were first downregulated than upregulated or vice versa (Table 1). Specifically, four DEGs were first downregulated at 6 hpt then upregulated at 24 hpt. These genes encoded JA-biosynthesis, reactive oxygen species (ROS) accumulation, aging, and wounding responses. Likewise, eight DEGs were first upregulated at 6 hpt then downregulated at 24 hpt. These genes were involved in response to iron ion (Fe) starvation, ABS accumulation, signaling, defense regulation, and responses to biotic/abiotic stress (Table 1). The expression patterns of shared DEGs at the two sampling points suggest modulation of crosstalk of SA, JA, and ABA pathways by SSG.

DEGs encoding stress responses accounted for the greatest portion of BP terms at both time points (Figure 2 and Figure 3a). However, the majority of DEGs were not present at the same time. To understand their roles in defense priming, the upregulated DEGs with FDR less than 0.001 were searched for their GO function. A total of 19 of the 62 and 25 of the 64 these DEGs were identified at 6 and 24 hpt, respectively. Among them, only AT3G24170 encoding cell redox homeostasis was shared at both time points. Those induced at 6 hpt were more and differed in functions from those at 24 hpt. The DEGs at 6 hpt encoded the processes involving SAR, the production of ABA, hydrogen peroxide (H_2_O_2_), Fe starvation response, and homeostasis, while those at 24 hpt encoded ABA production and ABA signaling and JA production/signaling. DEGs encoding abiotic and biotic responses were also more abundant by number and differed in functions at 6 hpt than 24 hpt. However, the DEGs encoding biotic response was missing, and those encoding other biological processes were much fewer at 6 hpt, indicating that SSG triggered abiotic responses followed by biotic responses related to SAR and SA production/signaling (Figure 4).

### 2.5. Differential Transcript Usage (DTU) with SSG and Altered Expression Patterns

Analysis of differential gene expression at the exon count level with DEXSeq showed that 308 and 245 transcripts were differentially expressed at 6 and 24 hpt, respectively (Appendix A). The result suggested that two or more isoforms were used in the regulation for some DEGs. More differential transcript usage (DTU) was detected at 6 hpt than 24 hpt at a higher significance level. At 6 hpt, 211, 84 and 13 transcripts expressed significantly at *p* = 0.05, *p* = 0.01 and *p* < 0.0001. A 24 hpt, 168, 63, and 13 transcripts expressed significantly at *p* = 0.05, *p* = 0.01 and *p* < 0.0001, suggesting that transcription by SSG was more active early.

Analysis of DEG and DTU with stageR revealed 9 and 11 genes with isoforms that showed statistically altered expression by SSG at 6 and 24 hpt, respectively (Appendix A). At 6 hpt (Appendix A), AT1G26440 had five isoforms, AT3G51840 and AT2G26910 had 4 isoforms, AT3G19960 had three isoforms and AT4G29010, AT4G32410, AT5G65780, AT5G46020, and AT4G37990 had two isoforms. Genes with different isoforms at 24 hpt (Appendix A) included AT5G17920, AT5G13740, and AT2G12400 with four isoforms, AT2G47070, AT2G27720, AT5G14060, and AT4G14210 with three isoforms, and AT4G13530, AT2G17220, AT2G2133, and AT3G16400 with two isoforms. However, not all the isoforms responded the same to SSG (Appendix A). Some isoforms expressed similarly in both mock and SSG treated plants, either induced (co-up) or not induced (co-down), although the expression levels may be different. Some isoforms were induced in the mock plant, but not the SSG treated plants or another way around. The occurrence of isoforms and various expression patterns of these genes suggested that SSG may induce alternative RNA splicing in plants to produce proteins that can simultaneously modulate multiple processes/functions. Only one out of 9 of these genes detected at 6 hpt encoded plant response to biotic stimulus, whereas 3 out of 11 detected for the same process at 24 hpt (Table 2), suggesting amplified plant defense priming from increased DEG diversity after treatment with SSG.

### 2.6. Biological Responses of Arabidopsis Col-0 and Mutants to SSG

SSG-treated Col-0 and mutant plants *eds*16-1, *npr*1-1, and *pad*4-1 did not differ from the controls until five days post-treatment (dpt) when leaves of *eds*16-1 became purplish or phosphorus (Pi) deficiency-like symptoms (Figure 5a). While SSG-treated Col-0, *npr*1-1, and *pad*4-1 remained no symptoms even at 10 dpt (Figure 5a), the leaves of *eds*16-1 turned darker after 5 dpt with SSG (Figure 5b). These results indicated that SSG neither affected wild-type *Arabidopsis* nor the *npr*1 plants with a deficiency in SA transducer NPR1 and *pad*4 plants with a defect in producing phytoalexin and SA biosynthesis/signaling. Instead, it affected *eds*16-1 plants with a deficiency in SA biosynthesis/signaling mediated by EDS16.

To understand how SSG may affect pigmentation in the plant, we examined the DEGs encoding chloroplasts where photosynthesis and pigmentation occur. Chloroplasts are the most encoded cellular components (Figure 3b and Table 1). Seven AGI loci involving activities in chloroplasts were identified in 23 shared DEGs at 6 and 24 hpt. For instance, AT3G06510 (SFR2, SENSITIVE TO FREEZING 2) was consistently downregulated at 6 and 24 hpt. AT4G38510 (ATVAB2, V-ATPASE B SUBUNIT 2, VAB2) was consistently upregulated at both time points. AT4G35770 (SEN1, Senescence-associated gene), together with AT1G52400, AT2G14247, AT5G14740, AT5G57350, was downregulated at 6 hpt but upregulated at 24 hpt. Purple pigmentation in leaves has been correlated with Pi deficiency [34]. Among these DEGs, AT4G35770 is related to Pi deficiency, which encodes a senescence-associated gene [35]. Yet, the expression of AT4G35770 did not cause Pi deficiency symptom on *Col*-0 (Figure 5a), suggesting disappearance of Pi symptom requires EDS16.

## 3. Discussion

This study with *Arabidopsis* transcriptome supported the hypothesis that SSG extended plant protection by inducing plant defense priming. The leaf endophytic *Burkholderia* mediated defense priming is different from the rhizobacteria that induce ISR, typically JA/ET pathways without activation of PR genes [6], and phyllosphere commensals that trigger defense priming through SA and JA pathways and upregulating PR genes mediated [18] in many ways. 

First, the priming involves a unique SA pathway. Detection of DEGs encoding SAR (AT1G55490 at 6 hpt, AT5G60600 at 24 hpt) (Figure 4) suggests PR gene regulation and SA usage in priming. Yet, none of the reported PR genes in the *Arabidopsis* plants colonized by phyllosphere commensals were detected as DEGs, suggesting that the SAR is a result of other PR genes. ACD6 (AT4G14400) is perhaps one of them, which as a DEG consistently upregulated at 6 and 24 hpt (Table 1). ACD6 encodes a transmembrane protein with intracellular ankyrin repeats and positively regulates cell death and defense, acting in part via SA and the SA transducer NPR1 [36,37,38] and is a critical component in the defense response against a broad spectrum of pathogens [39]. Although NPR1 was not a DEG, AT4G02520 was detected among the very significantly upregulated DEGs at 24 hpt (Figure 4). AT4G02520 may act similar to NPR1 since it is a binding protein of SA and camalexin and a glutathione transferase [40]. Camalexin and glucosinolates mediated SA pathway has been shown in defense priming induced by rhizobacterium [19], and the resulting defense is effective as SAR [41]. Therefore, AT4G14400 and AT4G02520 may be two important SA pathway regulators in the defense priming triggered by SSG.

Next, the priming also involves the JA pathway. AT4G35770 (SEN1) encoding JA response was not induced in Col-0 at 6 hpt but 24 hpt. *SEN*1 is a senescence-associated gene that is regulated by plant defense response linked signals and strongly induced by phosphate starvation [35,42]. The induction of *SEN*1 in *Col*-0 indicates that SSG regulates JA production and induces Pi deficiency in a delayed mode that may contribute to symptomless Pi deficiency on Col-0 (Figure 5a). *SEN*1 activation requires both SA and JA pathways involving the functions of EDS5, NPR1, and JAR1 [42]. EDS16 is an isochorismate synthase belonging to EDSs that are used by the TIR-NB-LRR class of R genes to promote system acquired resistance (SAR) [43,44,45]. It is not clear how SSG may induce *SEN*1 in the absence of DESs but based on the Pi-deficiency symptom on *eds*16-1. One possibility is that *SEN*1 is induced immediately after treatment, leading to enhanced sensitivity to JA/ET signaling and extensive Pi deficiency in *eds*16-1 due to lack of EDS16 or SA. Aging or built-up sugar happens under low Pi availability, leading to changes in leaf pigmentation [34,46]. *SEN*1 has been associated with the dark-inducible (DIN) gene that participates in senescence and pathogen invasion-related cellular events [47,48]. However, it is unclear whether it has any linkages to the JASMONATEZIM-domain (JAZ) family that play a key role in regulation between plant growth in response to light and defense against necrotrophic pathogens and insect herbivores [49,50,51].

Furthermore, SSG may trigger defense priming through regulating abscisic acid (ABA) biosynthesis and signaling. ABA as an antagonist of SA has a negative effect on plant-pathogen resistance. However, it plays a vital role in compatible mutualistic interactions such as mycorrhizae and rhizosphere bacteria with plants and can be suppressed at immune response [21]. The latter is evident in Col-0 plants treated with SSG. First, most of the DEGs encoding ABA response or signaling were consistently downregulated or became downregulated after an early upregulation at 6 hpt (Table 1). Second, DEGs such as AT3G07160 (BETA-GALACTOSIDASE 4, BGAL4) (Figure 4) encoding callose deposition in cell walls [52] were also upregulated at 6 hpt (Figure 4). Since ABA suppression can result in callose deposition enhancement providing an additional layer of protection during ISR [6], this upregulation of BGA4 may result from ABA suppression. On the other hand, ABA suppression by SSG is not consistent. Many DEGs related to ABA production and signaling were upregulated but mainly dependent on the lead time of treatment (Figure 4). DEGs encoding ABA production, including AT1G52400, AT1G22930, and AT2G05520, were mostly induced at 6 hpt, while DEGs encoding ABA signaling was primarily induced at 24 hpt (Figure 4). Although it is unclear how these DEGs led the priming, they may have dual roles. ABA can induce stomatal closure to prevent pathogens from entry [53]. It can also protect plants from pathogen infection through a distinct signaling pathway [31]. Thus, these DEGs for ABA production are likely used for priming for stomatal closure, augmenting structure barriers. By contrast, those for ABA signaling may be used in defense signaling crosstalk. JA/ABA has been identified as a positive pathway for resistance against insects [54]. In this pathway, ABA co-regulates the transcription factors MYCs in branches of the jasmonic acid (JA) signaling pathway. Although there were no DEGs related to MYCs that were detected, upregulation of SEN1 encoding response to endogenous JA suggests an operation of the JA/ABA pathway. SSG modulating the defense signaling crosstalk among SA, JA, and ABA may result in priming for enhanced defense against upcoming pathogens, although the molecular function of the crosstalk and how the crosstalk acts when the treated plants challenge pathogens or other stimuli encountered remains unclear. Endophytic *Actinobacteria*, a distinct bacterial group from *Burkholderia*, has been shown to induce priming for SA or JA/ET pathways for challenging upcoming fungal or bacterial pathogens [55]. SSG likely shares this ability and has a capacity to allow priming additional defense hormone signaling pathways, which explains why moderate resistance is in common in almost all the test plants treated with SSG at a long lead time before inoculation with pathogens [26,28].

Apart from triggering SA, JA/ET, and ABA-dependent priming, SSG also induces other priming mechanisms, such as ET and Fe deficiency response and regulation. Beneficial microbes-induced ISR overlaps with plant Fe deficiency response [56]. Fe deficiency response can result from the Pi deficiency-induced JA pathway through signal crosstalk [46,57]. It can also result from ethylene (ET) reduction that can be accomplished by the ACC deaminase-producing bacteria such as SSG [14,30]. Indeed, DGEs that encode Fe response to starvation (AT1G47395, AT1G47400), Fe homeostasis (AT2G41240), and response to endogenous ET (AT2G05520) were among those very significant DEGs at 6 hpt (Figure 4). This suggests that SSG stimulates ET production and the Fe-deficiency in the plants shortly after its application. ET can play a dual role in activating Fe deficiency responses and the onset of ISR [56]. AT1G47395 and AT1G47400 belong to FIT, the master regulator of the Fe acquisition gene network [58].

In *Arabidopsis*, several molecular markers have been shown to be useful for detecting the primed state [7]. Chromatin modification and activation of mitogen-activated protein kinases (MAPKs) based transcription factors have been proposed as other possible mechanisms of defense priming [7]. SSG may also be an inducer of such priming. AT1G57820, namely VIM1, is the SET- and RING-Associated domain methylcytosine-binding protein and H3K9 methyltransferase. It encodes chromatin organization and DNA cytosine methylation [59]. Chromatin modification can result in faster and stronger transcription of defense genes and memory priming for future challenges [60,61,62]. VIM1 expression was first downregulated (6 hpt) and then upregulated (24 hpt) (Table 1), suggesting priming for enhanced defense did not start until 24 hpt. Further investigation is warranted to determine whether the expression of VIM1 might have contributed to SSG long-lasting plant protection from pathogens [26,28]. MAPKs activation-related DEGs such as AT4G09570 (CPK4) and AT5G01810 (CIPK15) are also DGEs identified at 24 hpt (Figure 4). Both genes have functions in ABA signaling [63,64] thus they may serve as transcription factors in ABA pathway priming, although it is unknown whether they may act the same as MPK3/6 to enhance defense gene expression.

A few DEGs are interesting due to their functions in stress, biotic and abiotic responses, and relations to defense priming. For example, AT3G59930 encoding a defensin-such as protein, DEFL, and AT4G37990 ELI3-2 Defensins (Table 2) have recently been shown to confer broad-spectrum resistance to pathogens in crops [65]. DEFL functions at iron and zinc homeostasis and stress response [66]. A few other DEGs with multiple isoforms are also worth noting as mechanisms of defense priming (Table 2). These DEGs are AT4G37990 (ELI3-2) that is associated with a new type of R gene RPM1 for SAR against bacteria [67], AT3G24170 encoding cell redox homeostasis that is a critical scavenging and antioxidant machinery to detoxify harmful ROS [68], AT5G13740 encoding response to nematode, response to zinc ion, zinc ion homeostasis [69], AT2G12400 encoding ABA-related transcription factors [70], and AT3G16400 with roles of the glucosinolate-myrosinase system in plant defense responses [71].

## 4. Materials and Methods

### 4.1. The Plant Growth Conditions

*Arabidopsis thaliana* (Col-0) and SA-related mutants (all in the Col-0 background), *eds*16-1, *npr*1-1, and *pad*4-1 [72] were used. Briefly, *eds*16-1 is a mutant of isochorismate synthase; *npr*1-1 is a mutant of SA receptor protein NPR1; *pad*4-1 is a mutant of PhytoAlexin deficient 4 gene. Col-0 or each mutant was seeded in 6 9-cm pots with autoclaved Miracle Gro Moisture Control Potting Mix: Miracle Gro^®^ perlite (Scotts Mira-Gro Company, Marysville, OH, USA) at a 2:1 ratio. The plants were grown in a growth chamber at 25/20 °C with a 12/12 h light/dark cycle, watered as needed, and fertilized in the third week with 20-20-20 liquid fertilizer (Scotts, Marysville, OH, USA). Four-week-old plants were used for treatment. Plants after the treatment were grown under the same conditions and watered as needed.

### 4.2. Preparation of SSG Inoculum

SSG strain stored at −80 °C was recovered on the potato dextrose agar (PDA) after incubation at 28 °C for 48 h. A single colony of the recovered culture was then streaked on nutrient agar (NB) in 100 cm plates and sub-cultured for another 48 h. For cell suspension preparation, a dish of the subculture was scraped and mixed with 100 mL Phosphate-Buffered Saline (PBS) to have a concentration between 10^8^ and 10^9^ CFU mL^−1^. The concentration of the cell suspension presented as CFU was determined by plating 100 µL on PDA, as previously described [73].

### 4.3. Plant Treatment

Plants were cover sprayed with SSG suspension in PBS at 10^8^ to 10^9^ CFU mL^−1^ or PBS alone (for control plants) at 15 mL per pot. Each treatment had 3 replicates with 3 pots per replicate, and treated plants were arranged in a randomized complete block design in a moist storage plastic box with some water on the bottom. The box was lidded then placed in a growth chamber until the first sampling.

### 4.4. Analysis of the Effect of SSG on Plant Gene Expression

#### 4.4.1. Plant Sampling

Five Col-0 plants were sampled from each replicate pot at 6- and 24-h post-treatment (hpt), rinsed twice in 2 L distilled water, and once in 1 L sterilized distilled water (SDW). All the plants of 3 replicates from the same treatment in an experiment were pooled as a biological replicate and ground in liquid nitrogen and placed in 50 mL tubes and stored at −80 °C for future analysis.

#### 4.4.2. mRNA Extraction

Total RNA was first extracted from the 3 experiments for 3 biological replicates of each treatment at 6- and 24-hpt. For each of the 12 samples, 1 mL of the frozen plant powder was used to extract RNA by mixing with 5 mL TRIzol (Invitrogen, Waltham, MA, USA) in a 15 mL tube as instructed by the manufacturer. For mRNA or polyA+ RNA extraction, 250 µg of the total RNA was used for each sample, and the extraction was conducted with the mRNA Isolation Kit (Roche Applied Science, Penaberg, Germany). Both the total RNA and mRNA were quantified using the QuantusTM Fluorometer with QuantiFluro^®^ RNA System (Promega, Madison, WI, USA).

#### 4.4.3. Sequencing cDNA Proxy

The PCR-cDNA kit and barcoding protocol (PCB_9092_v109_revB_10Oct2019) of Oxford Nanopore Technologies (ONT) were used for sample cDNA generation, barcoding, and sequencing (ONT, Cambridge, UK). A total of 12 samples, including 3 biological replicates of 2 treatments, the control and SSG, and 2-time points, 6- and 24-hpt, were sequenced. For each sample, 2 ng of the polyA+ RNA was used at reverse transcription and strand-switching, 5 µL of the reverse-transcribed RNA was used for selecting for full-length transcripts and barcoding. For library preparation and sequencing, 12.5 ng of each of the amplified cDNA barcoded samples were pooled to obtain total cDNA of 100 fmol after DNA quantification using the QuantusTM Fluorometer with QuantiFluro^®^ dsDNA System (Promega, Madison, WI, USA) and size estimation. The size estimation was based on the average of transcript sizes of Arabidopsis, which was approximately 2 kb.

#### 4.4.4. Differential Gene Expression Analysis

cDNA fastq sequence data of the paired samples of 2 treatments (e.g., SSG/the control) at 6 and 24 dpt were analyzed through pipeline-transcriptome-de (https://github.com/nanoporetech/pipeline-transcriptome-de (accessed on 17 April 2021)) by following the ONT community bioinformatics tutorial “Using cDNA sequence collections for differential transcript usage” (https://community.nanoporetech.com/knowledge/bioinformatics/tutorials, accessed on 15 April 2021). The differential transcript usage (DTU) and differential gene expression (DGE) of paired samples were analyzed for the identification of genes and transcripts that appear differentially abundant between the treatments with and without SSG. AtRTD2 (Arabidopsis Thaliana Reference Transcript Dataset 2 [74], including AtRTD2_19April2016.fa and AtRTD2_19April2016.gtf, were used (https://ics.hutton.ac.uk/atRTD/, accessed on 17 April 2021). The workflow used tools, including Minimap2 for mapping long sequence reads to the reference transcriptome, salmon for counting reads that map to a transcript, and various R packages (edgeR, DEXSeq, DRIMSeq, and stageR) for the statistical analysis of the gene and transcript associated mapping information.

#### 4.4.5. Gene Ontology (GO) Enrichment Analysis and Functional Categorization of Regulated Genes

The significant differentially expressed genes (DEG) in the 6 h and 24 h were, respectively, selected as input for the via an online toolkit: The plant gene set enrichment analysis [75]. This web server compiled many GO terms for *Arabidopsis thaliana*. The GO gene sets contained 1,018,369 entries with the plant genes identified in biological process (BP), cell component (CC), molecular function (MF). Fisher exact was used for statistical testing. Benjamini–Hochberg post-hoc test was used to adjust multi-test *p*-values. The false discovery rate (FDR) was set at 5%. Pheatmap [76] was used to visualize the overlapping genes across the significant GO terms. The Reduce Visualize Gene Ontology (Revigo) web server [77] was used to calculate the similarity of the significant GO terms and produce a visualization of the BP term classification. SimRel method was selected for the semantic similarity measure. *A. thaliana* was used as a reference organism.

The differentially expressed genes at the 2 sampling points were categorized through The Arabidopsis Information Resource (TAIR) using GO annotation search (https://www.arabidopsis.org/tools/bulk/go/index.jsp, accessed on 9 December 2021). The search results were used for functional annotation to understand the role of SSG in plant defense priming.

### 4.5. Biological Responses of Arabidopsis to SSG

All treated plants in each experiment, including the control, were examined for physical changes at 1-, 3-, 5-, 7-, and 10-days post-treatment (dpt).

## 5. Conclusions

Analyzing *Arabidopsis* plants inoculated with the cell suspension of a burkholderial biocontrol agent SSG revealed that SSG is a unique ISR inducer different from rhizobacteria via JA/ET, and phyllosphere commensals via common SA and JA pathways. SSG uses the transmembrane protein ACD6 and NPR1 -mediated SA pathways and SEN1 mediated JA pathway that regulates JA production and induces Pi deficiency response. SSG also regulates ABA biosynthesis and signaling, leading to callose deposition enhancement, defense signaling crosstalk, and stomatal closure modulation. Additionally, SSG uses other molecular mechanisms for plant defense priming. These include inducing response to Fe deficiency, regulating ET, modifying chromatin through VIM1, activating MAPKs (CPK4 and CIPK15), and regulating DEFL for iron and zinc homeostasis and stress, biotic and abiotic responses. Thus, SSG is not only a pathogen suppressor and plant growth promoter [26,27,28] but also a trigger of plant defense priming with diverse mechanisms. The priming triggered by SSG can occur shortly after the application of the cell suspension when the biocontrol agent is still alive, directly interacting with encountered pathogens. As a result, the priming leads to broad-spectrum and long-lasting moderate plant resistance even though SSG cells die-off, as demonstrated previously [26,28,29].

## Figures and Tables

**Figure 1 ijms-23-03151-f001:**
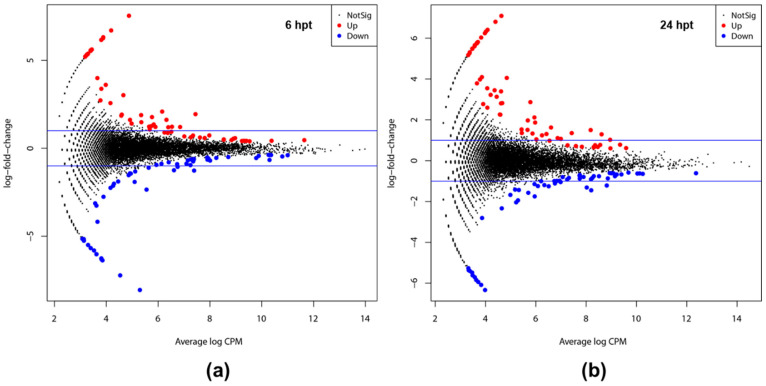
MA plot based on the EdgeR analysis showing differences in measurements between SSG-treated and control plants at 6 (**a**) and 24 hpt (**b**). M, log-fold-change (−log_2_ FC) (ratio of gene expression calculated between the control and SSG treated plants). A, Average log CPM, or log2 (counts per million abundances, CPM). Schemes follow the same formatting.

**Figure 2 ijms-23-03151-f002:**
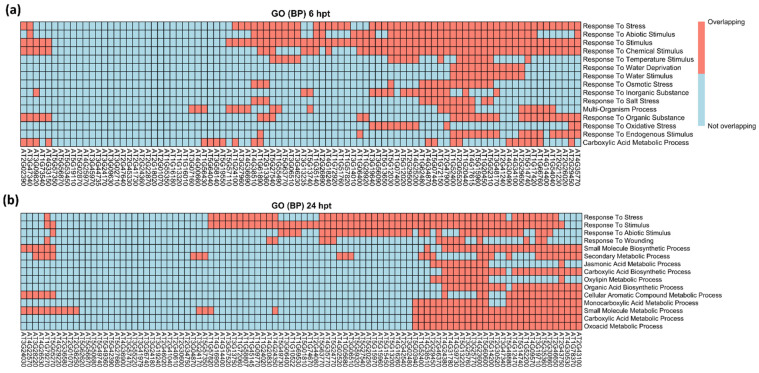
Heatmap of the differentially expressed genes encoding the top 15 significant GO terms. (**a**) Significant BP terms at 6 hpt. (**b**) Significant BP terms at 24 hpt. Overlapping genes were orange and not overlapping genes were light blue. Schemes follow the same formatting.

**Figure 3 ijms-23-03151-f003:**
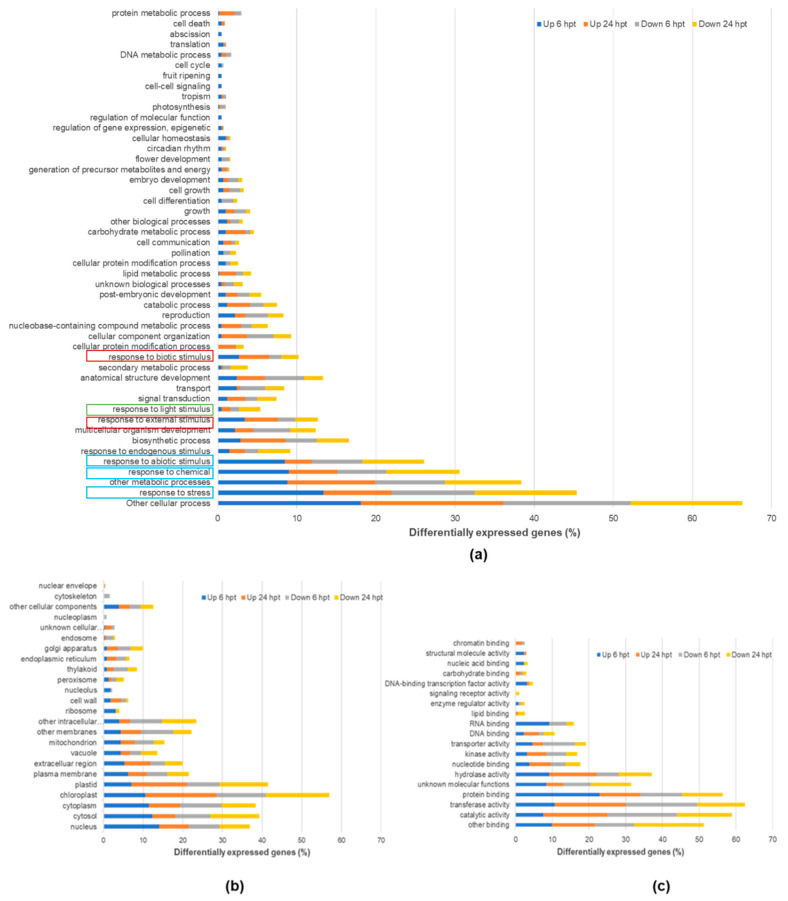
Percentage of differentially expressed genes (DEGs) by SSG for BP (**a**), CC (**b**), and MF terms (**c**) at 6 and 24 hpt. The expressed genes were determined with EdgeR analysis and separated by their regulation type, down or up. GO terms and functional categorization of the genes are generated through TAIR (The Arabidopsis Information Resource). The GO term of interest in the same box of color depicts a similar pattern. Schemes follow the same formatting.

**Figure 4 ijms-23-03151-f004:**
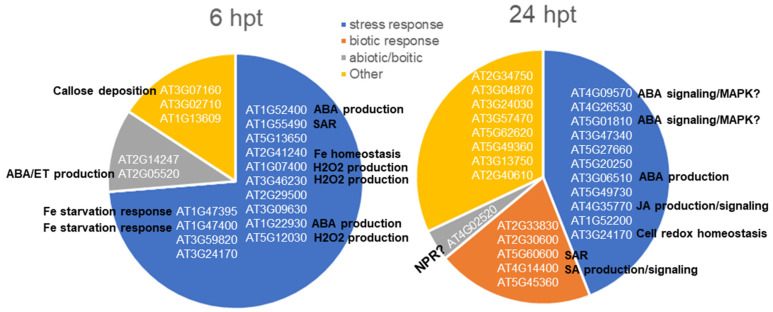
Upregulated DEGs in *Arabidopsis* Col-0 at 6 and 24 hpt with SSG at FDR < 0.001 determined by EdgeR analysis. Functional categorization of the genes is based on GO annotation on TAIR (The Arabidopsis Information Resource). The DEGs functions involving defense priming are bold.

**Figure 5 ijms-23-03151-f005:**
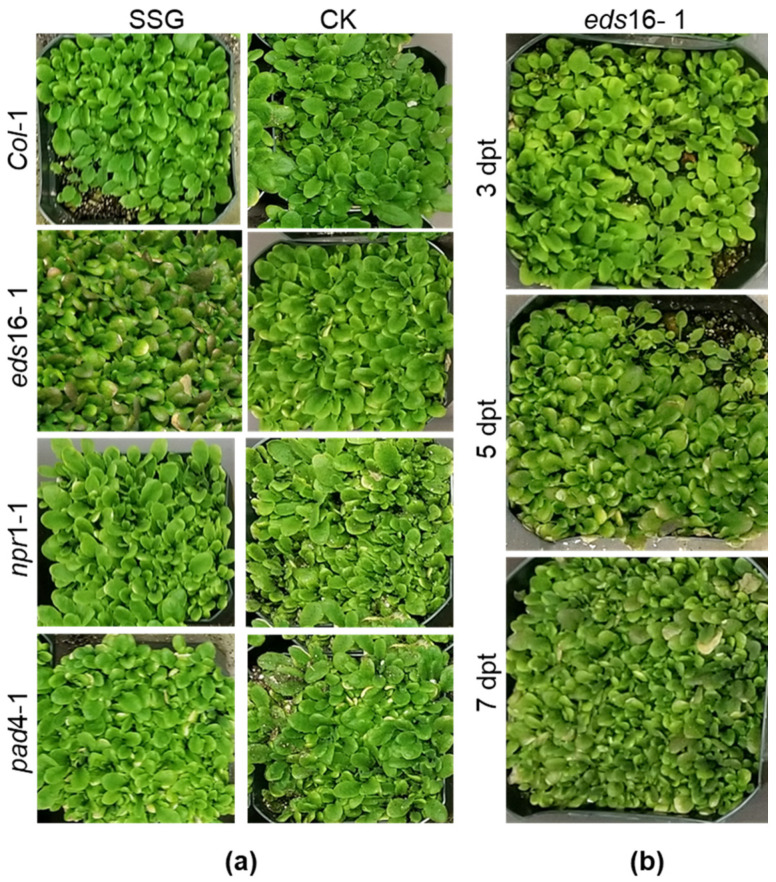
Responses of *Arabidopsis* to *Burkholderia* sp. SSG. Four weeks-old plants were sprayed with SSG at 10^8^ to 10^9^ CFU mL^−1^ PBS or the control CK, PBS alone. (**a**) The morphology of Col-0 and its mutants at 10 days post-treatment (dpt). (**b**) Symptom development on *eds*16-1 after SSG spray.

**Table 1 ijms-23-03151-t001:** Shared differentially expressed genes (DEGs) at 6 and 24 hpt with SSG.

AGI ^W^	Regulation	Log_2_ FC ^X^				GO Biological Process ^Z^
		6 hpt ^Y^		24 hpt		
AT2G01070	Down at 6 hpt/down at 24 hpt	−2.8	*	−2.8	*	Transport
AT2G29450		−0.5	*	−0.8	**	Response to oxidative stress, glutathione and toxin catabolic process
AT4G38510		−6.3	***	−5.7	**	Actin filament capping, bundle assembly, and depolymerization; ATP metabolic processes
AT5G15960		−0.7	*	−1.0	**	Response to ABA, Response to osmotic stress, cold, drought, light signaling
AT5G27540		−1.4	*	−5.8	***	Embryo development, mitochondrion organization, ending in seed dormancy, regulation of mitochondrion, pollen tube growth
AT5G63770		−1.9	*	−1.7	*	Leaf, root development, response to wounding and cold
AT3G06510	Up at 6 hpt/up at 24 hpt	5.6	**	6.8	***	Response to freezing, cold; carbohydrate metabolic proc
AT3G24170		6.2	***	6.4	***	Cell redox homeostasis, glutathione metabolic process
AT4G14400		3.4	**	3.4	***	**Defense response** to virus, bacteria, fungi, and oomycetes; cell death, Response to SA, light, freezing; SA signal regulation, protein ubiquitination
AT5G56870		5.4	*	5.7	**	Organic substance catabolic, immune system proc, **defense response** to fungi, response to bacteria, hypoxia, oxidative stress, wounding, light, water deprivation; organic cyclic compound, leaf senescence
AT2G17200		5.5	**	5.2	*	Epidermal cell differentiation, **defense response**, ubiquitin-dependent protein catabolic proc, organelle organization, response to lipid
AT1G57820	Down at 6 hpt/up at 24 hpt	−6.3	***	5.7	**	Chromatin organization, DNA methylation on cytosine, protein ubiquitination, cell division, heterochromatin assembly
AT3G47340		−1.3	***	2.1	***	Asparagine biosynthetic process, response to sucrose, fructose, and glucose; darkness, cellular amino acid catabolic proc
AT3G59930		−0.7	***	0.7	*	Response to light
AT4G35770		−0.6	***	1.3	***	Response to ROS, aging, JA, wounding
AT1G13609	Up at 6 hpt/down at 24 hpt	2.1	***	−1.4	*	Unknown
AT1G47395		1.9	***	−1.3	***	Positive regulation of iron ion transport, response to iron ion starvation
AT1G47400		1.8	***	−2.0	***	Positive regulation of iron ion transport, response to iron ion starvation
AT1G52400		0.7	***	−0.8	***	**Defense response** to fungi; water deprivation, insect, glucosinolate catabolic proc, protein polymerization, ABA proc; ABA and salt response; stomatal movement, regulation of ABA signaling
AT2G14247		1.6	***	−1.2	**	Glucosinolate catabolic proc, regulation of biological quality Cellular response to hypoxia
AT4G25670		3.8	*	−3.1	*	Response to cold, organonitrogen compound, water deprivation, wounding; regulation of defense, defense response to biotic stimulus, ABA signaling
AT5G14740		0.6	*	−0.5	*	Carbon utilization
AT5G57350		1.4	*	−6.3	***	Proton and ion transmembrane transport, intracellular pH regulation

^W^ Gene identifier by Arabidopsis Genome Initiative (AGI). ^X^ the log_2_ ratio of gene expression between the control and SSG treated plants calculated with the edgeR. ^Y^ hpt = hours post treatment, Asterisks denote difference significance of gene expression with SSG measured with false discovery rate (FDR) * < 0.05, ** < 0.01, *** < 0.001. ^Z^ GO terms generated through The Arabidopsis Information Resource (TAIR), representing biological process, cellular component, and molecular function.

**Table 2 ijms-23-03151-t002:** Functional categorization of biotic stimulus-response DEGs with isoforms at 6 and 24 hpt.

AGI	Transcript ID (Expression Pattern) *	GO Biological Process	GO Molecular Function	GO Cellular Component
**6 hpt** AT4G37990 (isoforms = 2, *p* = 0.0246)	P1 (down) ID2 (up)	other cellular processes **response to biotic stimulus** **response to external stimulus** other metabolic processes cell death **response to stress** biosynthetic process secondary metabolic process	catalytic activity other binding	cytoplasm plasma membrane
**24 hpt** AT5G13740 (isoforms = 4, *p* = 5.32E-13)	P1 (co-up) ID8 (down) ID4 (up) ID10 (down)	other cellular processes **response to biotic stimulus** transport **response to external stimulus** **response to chemical** other biological processes	transporter activity	vacuole plasma membrane cytoplasm other membranes mitochondrion
**24 hpt** AT2G12400 (isoforms = 4, *p* = 0.0244)	P1 (up) JS2 (co-down?) JS1 (co-down?) ID3 (co-down?)	other cellular processes **response to chemical** other metabolic processes **response to abiotic stimulus** **response to stress** growth anatomical structure development **response to biotic stimulus** **response to external stimulus** lipid metabolic process cell growth response to endogenous stimulus signal transduction other biological processes	unknown molecular functions	other cellular components
**24 hpt** AT3G16400 (isoforms = 2, *p* = 0.0391)	P1 (co-up) ID5 (up)	other cellular processes other metabolic processes secondary metabolic process **response to biotic stimulus** **response to external stimulus** biosynthetic process catabolic process	RNA binding enzyme regulator activity	extracellular region nucleus cytoplasm cytosol

Functional categorization is based on TAIR (The Arabidopsis Information Resource). AGI: Gene identifier by Arabidopsis Genome Initiative. * co-up or co-down: detected or undetectable transcripts were in most replicates of both CK and SSG; up or down: detected or undetectable transcripts were only in most replicates of SSG; co-down?: uncertainty of the pattern due to presence of the detected or undetected transcripts in only one replicate of SSG samples. GO BP terms related to responses to abiotic and biotic stimuli are bold.

## Data Availability

The cDNA sequences of Arabidopsis thaliana Col-0 of two treatments (sprayed with *Burkholderia* sp. SSG cell suspension and mock) in three biological experiments are deposited at DDB/ENA/GenBank under BioProject PRJNA796315, accession: SAMN24843699-SAMN2484370. Differential expression data are included in Appendix A.

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
