# Peer review of "cDNA Transcriptome of Arabidopsis Reveals Various Defense Priming Induced by a Broad-Spectrum Biocontrol Agent Burkholderia sp. SSG"

_ijms, 2022, doi:10.3390/ijms23063151_

Round 1
Reviewer 1 Report
The manuscript "cDNA transcriptome of Arabidopsis reveals various defense 2 priming induced by a broad-spectrum biocontrol agent 3 Burkholderia sp. SSG" is well written, covering all the relevant aspects of the topic.
There are minor writing errors such as use consistent units (for time use one of h or hr or hour). Use one unit throughout. Minor sentencing errors that should be easily fixed in re-reading of manuscript.
I am not sure, if any submissions to NCBI database are needed for this manuscript. Kindly look into it.
Reviewer 2 Report
Dear Authors
The current manuscript entitled “cDNA transcriptome of Arabidopsis reveals various defense priming induced by a broad-spectrum biocontrol agent Burkholderia sp. SSG” demonstrated and suggested that SSG is an induced systemic resistance (ISR) trigger conferring plant protection upon pathogen encounter. The present work has been very well planned, executed and presented nicely. Although there are some queries and further opportunities to improve the manuscript.
- For cell suspension preparation, a dish of the subculture was suspended in 100 ml Phosphate-Buffered Saline (PBS) to have a concentration at 108 to 109 CFU mL-1. Please explain if colony from subculture was grown in PBS, or the scraped colonies were just mixed in PBS? And the concentration was 108 or 109?
- Humidity in the growth chamber during the experiment?
- A small concluding remark may be included to highlight the key finding of the study.
